# Nanoarchitectonics for Ultrathin Gold Films Deposited on Collagen Fabric by High-Power Impulse Magnetron Sputtering

**DOI:** 10.3390/nano12101627

**Published:** 2022-05-10

**Authors:** Sheng-Yang Huang, Ping-Yen Hsieh, Chi-Jen Chung, Chia-Man Chou, Ju-Liang He

**Affiliations:** 1Department of Materials Science and Engineering, Feng Chia University, 100, Wenhwa Rd., Seatwen District, Taichung 40724, Taiwan; drugholic@vghtc.gov.tw (S.-Y.H.); pyhsieh@fcu.edu.tw (P.-Y.H.); 2Department of Surgery, Taichung Veterans General Hospital, 1650, Sec. 4, Taiwan Boulevard, Seatwen District, Taichung 40705, Taiwan; 3Department of Medicine, National Yang-Ming University, 155, Sec.2, Linong Street, Beitou District, Taipei 11221, Taiwan; 4Department of Dental Technology and Materials Science, Central Taiwan University of Science and Technology, 666, Buzih Rd., Beitun District, Taichung 40601, Taiwan; cjchung@seed.net.tw; 5Institute of Plasma, Feng Chia University, 100, Wenhwa Rd., Seatwen District, Taichung 40724, Taiwan; jlhe@fcu.edu.tw

**Keywords:** collagen, high-power impulse magnetron sputtering (HiPIMS), ultrathin gold film (UTGF)

## Abstract

Gold nanoparticles conjugated with collagen molecules and fibers have been proven to improve structure strength, water and enzyme degradation resistance, cell attachment, cell proliferation, and skin wound healing. In this study, high-power impulse magnetron sputtering (HiPIMS) was used to deposit ultrathin gold films (UTGF) and discontinuous island structures on type I collagen substrates. A long turn-off time of duty cycle and low chamber temperature of HiPIMS maintained substrate morphology. Increasing the deposition time from 6 s to 30 s elevated the substrate surface coverage by UTGF up to 91.79%, as observed by a field emission scanning electron microscope. X-ray diffractometry analysis revealed signature low and wide peaks for Au (111). The important surface functional groups and signature peaks of collagen substrate remained unchanged according to Fourier transform infrared spectroscopy results. Multi-peak curve fitting of the Amide I spectrum revealed the non-changed protein secondary structure of type I collagen, which mainly consists of α-helix. Atomic force microscopy observation showed that the roughness average value shifted from 1.74 to 4.17 nm by increasing the deposition time from 13 s to 77 s. The uneven surface of collagen substrate made quantification of thin film thickness by AFM difficult. Instead, UTGF thickness was measured using simultaneously deposited glass specimens placed in an HiPIMS chamber with collagen substrates. Film thickness was 3.99 and 10.37 nm at deposition times of 13 and 77 s, respectively. X-ray photoelectron spectroscopy showed preserved substrate elements on the surface. Surface water contact angle measurement revealed the same temporary hydrophobic behavior before water absorption via exposed collagen substrates, regardless of deposition time. In conclusion, HiPIMS is an effective method to deposit UTGF on biomedical materials such as collagen without damaging valuable substrates. The composition of two materials could be further used for biomedical purposes with preserved functions of UTGF and collagen.

## 1. Introduction

Currently, collagen is one of the main materials for commercially available wound dressings. Most collagen comes from natural sources, such as bovine, porcine, human blood, and marine organisms, due to its relatively low cost [1]. If pure collagen is not treated by cross-linking agents, then it may be easily hydrolyzed or degraded because its tissue strength is weaker than that of the skin and the surrounding environment. However, most cross-linking agents are chemical agents, such as polyethylene oxide, genipin, glutaraldehyde, 1-(3-dimethylaminopropyl)-3-ethylcarbodiimide hydrochloride (EDC), and EDC plus N-hydroxysulfosuccinimide [2,3,4]. Concerns about toxicity residues exist due to redox chemical reactions and environmental pollutants generated during the preparation process.

In tissue engineering applications, collagen often incorporates other functional additives, such as growth factors, polymer fibers, antibiotics, metal nanoparticles, and active cells [4,5,6,7], to provide different functions. For example, the addition of growth factors can promote the activation of stem cells, and the addition of antibiotics can increase the bactericidal function. Recently, gold nanoparticles (AuNPs) conjugated with collagen scaffold have been shown to improve stability and resistance against degradation [8], immobilization of peptides for easy incorporation [9], biocompatibility enhancement [10], and prevention of intraluminal thrombosis [11], which can be beneficial for bone growth, myocardium or vascular regeneration, and skin repair [4].

Among the conventional AuNP preparation processes, the physical method is more favored. Physical vapor deposition (PVD) can be divided roughly into evaporation and sputtering. Common evaporation techniques, such as molecular beam epitaxy (MBE), have been proven to produce metal nanoparticles by rapidly heating the target to manufacture evaporated molecules and to collect by condensation method. Metal nanoparticles could be further added in raw biomedical materials and shaped into desired configuration [10]. Different sizes and shapes of nanoparticles could be achieved by properly adjusting the evaporation time and power supply of vacuum evaporation [12]. However, direct surface modification of biomedical materials by MBE or pulsed laser ablation is difficult because of substrates breakdown in such an environment. In addition, MBE is an expensive and time-consuming process. Another PVD called electron beam (E-beam) evaporation has the problems of high working temperature, unstable film thickness, uncontrolled surface coverage, and poor thin film adhesion. E-beam is also expensive, and the difficulty of deposition control further restricts the direct application on biomedical materials. Generally, thin film quality is difficult to be controlled by evaporation and adhesion is less strong than sputtering. Collagen is prone to thermal damage thus common treatment with substrate heating should be avoided. The associated heating brought by the arriving ions accompanied by sputtering could increase substrate temperature. Additional substrate heating is a common, but not necessary, process of ion beam deposition, direct current sputtering, or radio frequency sputtering [13]. Unlike traditional sputtering, high-power impulse magnetron sputtering (HiPIMS) has higher peak power, higher target surface ionization, stronger magnetic field, and a longer turn-off time (T_off_) of duty cycle. HiPIMS can produce desired thin films and film adhesion without substrate heating or substrate bias [14]. The goal of this study is to deposit ultrathin gold films (UTGF). Thus, the whole processing time is assumed to be short. The impact on substrate properties, surface characteristics, microscopic structure, and temperature is low during HiPIMS deposition. A literature review on biomedical material modification by HiPIMS mainly focused on hard substrates, such as bone implants and dental implants, to increase hardness, corrosive resistance, and biocompatibility [15,16,17,18,19,20,21].

There have been many reference papers describing AuNPs for biomedical application [4,8,9,10,11]. However, the paucity of UTGF research for biomedical materials is the main motivation of the current study. Furthermore, no reports of UTGF deposition on collagen by HiPIMS have been published so far. Direct surface modification via the deposition of UTGF on biomedical substrates could preserve the properties of both materials. This study aims to deposit UTGF on type I collagen substrates, which is expected to promote cell adhesion, proliferation, and angiogenesis, as well as accelerate wound healing. The surface morphology, microstructures, and water contact angles (WCAs) were determined to explore the surface characteristics by tuning the process parameters. In this study, the research team mainly focused on the deposition, gross behavior, and surface characteristics of UTGF by HiPIMS as the foundation of further applications.

## 2. Materials and Methods

Two kinds of collagen substrates were used: substrate A (Sunmax Collagen Dental Membrane, Sunmax Biotechnology Co., Ltd., Tainan, Taiwan), made from highly purified porcine dermal type I collagen matrix, compressed to a semi-translucent membrane with a thickness of 0.01 cm; and substrate B (HealiAid^®^, Maxigen Biotech Inc., Taipei, Taiwan), made from bovine Achilles tendon-based type I collagen fabric and having a thickness of 0.25 cm, which is pliable, absorbable, designed for cleaning the skin for wound treatment, and has more water-absorbing capacity. To clarify, substrate A was a membranous type of materials, and the surface was more appropriate for surface roughness measurement and UTGF crystal structures observation to minimize confounding localization factors. However, the manufacturing process of substrate A involved high pressure and the natural fiber alignment of collagen was altered. For substrate B, the native fibrillar structure was preferred for tests of surface wettability and elemental chemical states. The collagen substrates were pretreated with argon plasma bombardment at a working pressure of 0.75 Pa for 5 min using a RF power supply set at power of 50 W to eliminate contamination on the substrate surface.

The schematic of the HiPIMS system is shown in Figure 1 and the process parameters for UTGF deposition on collagen substrates is in Table 1. A vacuum chamber with a single rectangular magnet was used for HiPIMS. Collagen substrates were fixed on the holder, and pure gold target (99.99%, 11.4 cm × 33.5 cm, purchased from Solar Applied Materials Technology Co., Tainan, Taiwan) was used for sputtering. The closest distance between the target and the rotating substrates was 6 cm, the substrate rotational speed was set as 20 RPM, and the power supply was provided by TruPlasma Highpulse 4002 (G2TRUMPF Hüttinger Sp. z o. o.). A high peak current density and duty cycle could damage the substrate by overheating. Based on a few preliminary test rounds, the peak current and peak power were maximized, and collagen substrate damage was prevented at the same time. The performance of UTGF on collagen according to deposition time was then studied. The deposition time was based on test rounds on other substrates with the desired UTGF thickness.

Surface and cross section morphology was observed by using a cold field emission scanning electron microscope (FE-SEM, Hitachi S-4800, Tokyo, Japan). Bare substrates were deposited with platinum for the FE-SEM study due to insulating nature, and UTGF-deposited specimens were studied without platinum film coating. Platinum deposition parameters were vacuum of 10 Pa, deposition time of 10 s, and current of 10 mA by a DC sputtering machine (Hitachi E-1010 Ion Sputter, Tokyo, Japan). The elemental composition of the specimen was determined using an energy-dispersive spectrometer (EDS, HORIBA EMAX400, Kyoto, Japan) installed on the FE-SEM (10,000 times magnification and accelerating voltage of 15 kV). The crystal structures of UTGF were determined using an X-ray diffractometer (XRD, Bruker D8 Discovery, Billerica, MA, USA) with Cu Kα radiation in the range of 2θ: 10–90°, grazing angle of 0.1°, step size of 2θ: 0.04° and a counting time of 0.6 s per step. The software used for phase identification was JADE Standard 2020 (Materials Data, Livermore, CA, USA). The surface topography of UTGF-deposited substrate A was studied by using atomic force microscopy (AFM, Bruker, Dimension Icon, Billerica, MA, USA) in tapping mode with a scanning area of 1 × 1 μm^2^ and demonstrated in reconstructed height images. Surface roughness values, including roughness average (Ra) and maximum roughness depth (Rmax), were measured.

Attenuated total reflection-Fourier transform infrared (ATR-FTIR) spectroscopy (Perkin Elmer, Frontier™ FT–IR Spectrometer, Waltham, MA, USA) with a resolution of 8 cm^−1^ and accumulated scanning of 16 times was used to analyze the functional groups and chemical bonds of the specimen surface. The studied wave number spectrum was 650–4000 cm^−1^ to cover the characteristic amide bands of collagen. The reported spectrum of type I collagen was used to confirm the purity of collagen substrates used for this study [22,23]. Peak changes of different wave numbers were evaluated and compared. The 1590–1720 cm^−1^ spectral interval, which includes amide I, was used to determine the secondary structure of collagen. Amide I band peaks were deconvoluted with a Lorentzian line shape function using OriginPro 9 (OriginLab Co., Northampton, MA, USA) software. The curve was then fitted with Gaussian band shapes to study the secondary structure of protein according to the reported method [24,25]. X-ray photoelectron spectroscopic (XPS) spectrum was collected by using an ULVAC-PHI, PHI 5000 VersaProbe/Scanning ESCA Microprobe system (Kanagawa, Japan) in the vacuum of 1 × 10^−9^ mbar with a monochromatized Al Kα X-ray source. Quantitative surface survey analysis and high-resolution scanning for Au, C, N, and O were performed. The X-ray beam size was 500 μm, and survey spectra were recorded with pass energy (PE) of 150 eV, step size of 1 eV, and dwell time of 50 ms, whereas high-energy resolution spectra were recorded with PE of 20 eV, step size of 0.07 eV, and dwell time of 100 ms. (The average number of sweeps was 5–30.) At least two points on the specimen surface were randomly selected as the scanning area to confirm the reproducibility and to obtain information on surface homogeneity. During analysis, the high-resolution spectra were deconvoluted using various Gaussian–Lorentzian components with the background subtracted in Shirley mode.

A First Ten Ångstroms 1000 (FTA 1000) WCA goniometer system (Newark, CA, USA) was employed to measure the WCA of bare collagen substrates and UTGF deposited on collagen substrates. Three sites of one specimen were randomly selected, and the mean WCA of the specimen was calculated for surface energy study. The morphology of the specimen was photographed after water absorption by collagen substrates.

## 3. Results and Discussion

### 3.1. Gross Appearance of Untreated and UTGF-Deposited Collagen Substrates

Both collagen substrates are sterilized in gamma irradiation for medical application by manufacturers. However, the mechanism of gamma irradiation is to break down bacterial deoxyribonucleic acid and inhibit bacterial division [26]. For the HiPIMS process, residual fragments of bacteria, impurities, or cross-linking agents used for collagen fabrication are unfavorable for thin film adhesion. Pretreatment of the collagen material surface before gold deposition in the HiPIMS chamber is therefore essential. The species of glow discharge generated by oxygen plasma are more reactive, especially for organic materials such as protein. Surface functional groups and chemical bonds are possibly changed after oxygen plasma treatment. Argon plasma treatment is therefore favored to remove surface contamination of the collagen substrates. After pretreatment with Ar plasma and initial trial of HiPIMS gold deposition on substrate A, the appearance of the specimens was photographed, as shown in Figure 2a. Bare substrate A is semilucent with a pale white color. UTGF-coated specimens of deposition times 13 and 26 s retained their characteristic transparency and revealed a dark gray color, but those of deposition times 51 and 77 s exhibited a light gold color. Bare substrate B is white to yellow with some wrinkles. To create feasible and functional biomedical composites, deposition time was adjusted as 6, 12, 18, 24, and 30 s for substrate B. The color of the specimens was orange to pink after 6 and 12 s, and light gray after 18, 24, and 30 s, as shown in Figure 2b. The colorful appearance revealed nanoscale gold on the surface due to specific surface plasmon resonance phenomenon [27]. AuNPs usually represent red or blue in color depending on the particle sizes due to localized surface plasmon resonance (LSPR) effect. The different electron oscillation modes on the surface of NPs and thin films created different appearances. The “golden” color of 51 and 77 s specimens was similar to bulk gold as the absorption of visible light by the thicker films. Collagen is vulnerable in high-temperature and water-containing environments [28,29,30]. Vacuum in HiPIMS is generally considered harmful for vulnerable materials such as collagen. However, some evidence revealed the safety and insignificant change of protein in low- and even high-vacuum conditions [31,32]. The extremely dehydrated environment achieved in vacuum is supposed to avoid protein degradation. Deformity of substrate A was observed in a drying oven at 40 °C after 40 min (photos not shown here). In contrast, both substrates A and B were grossly intact without shrinkage, melting, or perforation after HiPIMS treatment.

### 3.2. Surface Morphology and Microstructures of UTGF Deposited Specimens

SEM images of UTGF-coated substrates A and B of different deposition times from the top view and oblique cross-sectional view are shown in Figure 3. The SEM images of substrate A with deposition times 26 and 51 s (Figure 3a) revealed the preserved fibril structure of collagen, and the width of the fibrils was about 1–1.5 μm, which was consistent with the literature description [14,33]. A wavy appearance on the surfaces may be attributed to the overlapping of dark and light bands of collagen fibers. The surface morphology of deposition time (51 s) showed a flattening of the wavy surface, which suggested that a longer deposition time contributed to the higher cohesive force of AuNPs, resulting in continuous thin film formation. SEM images of substrate B with deposition times of 6, 12, 24, and 30 s, as shown in Figure 3b, also revealed preserved collagen fibers after UTGF deposition. The interfibrillar space (between the yellow lines) of bare substrate B was around 300 nm but decreased to 200 nm in the specimen of deposition time 30 s. Under the microscopic field, the inhomogeneity of the fibrillar structure inhibited the representativeness of interfibrillar space measurement. Compared with bare substrate B, the specimen of deposition time of 6 s showed AuNP deposition on the fibril surfaces, which were smaller than 10 nm. The specimen of deposition time 30 s showed a wavy appearance as type 1 collagen fibers, which indicated islands and discontinuous film growth of HiPIMS gold deposition between deposition times of 6 and 30 s. The growth pattern fitted well with the Volmer–Weber model rather than layer-by-layer growth theory in the early phase of thin film growth [34,35].

Bare substrate B has larger interfibrillar space, which benefits cellular contact, larger surface area ratio, and water absorption capability [14,33,36]. After surface modification by UTGF deposition, the exposed area of collagen decreased with the increasing deposition time, but continuous film formation was not observed due to the larger interfibrillar space, as shown in Figure 4. The SEM images of different deposition times and the ratios of collagen on surface to collagen at deeper levels are also shown in Figure 4 and processed with ImageJ software, version 1.53p (Rasband, W.S., ImageJ, U.S. National Institutes of Health, Bethesda, MD, USA). The selection of the substrate exposed surface area (ESA) was performed in grayscale mode before and after UTGF deposition. ESA percentage was calculated by using integral images [37,38]. More than half of the bare substrate B surface area was composed of gaps. ESA decreased as UTGF deposition time increased.

FEM-EDS analysis on collagen would be difficult at high temperatures due to collagen’s heat intolerance. Thus, scanning energy (kV) and electron beam bombardment time should be adjusted carefully [39,40]. The deposition of Au or Pt for bare collagen substrates would further influence the analysis results. Table 2 shows the results of the EDS surface element analysis of UTGF-coated substrates A of different deposition times. A linear increase in the Au atom percentage (0.10, 0.31, 0.72, and 0.98%) was observed as the deposition time increased (13, 26, 51, and 77 s). The C/N atomic ratio was between 3.09 and 3.48, which correlated with literature reports about collagen of natural origins [41]. The atom percentage of Au was smaller than 1% even on specimens of deposition time 77 s, which was influenced by the deeper detection depth from the surface by EDS. In consideration of the lower surface resolution of EDS, XPS was further used to study surface element analysis. The atom percentage of Au increased significantly by increasing deposition time (9.70, 10.80, 14.70, 31.50, and 44.30% from specimens of 6, 12, 24, 48, and 96 s). The results showed that sputtered Au ions only deposited on the surface of collagen substrates without deeper penetration via the porous structure or interfibrillar space [42]. The observation is related to the short deposition time and low peak power of HiPIMS deposition. The SEM image and Au (M series) mapping of UTGF-coated substrate B of deposition time 6 s are shown in Figure 5a. The homogenous distribution of Au on collagen fibers was clearly observed, which was not related to intermittent dark or light bands. HiPIMS Au deposition on the substrate materials with gaps was low. Surface modification by UTGF deposition would not affect the structure and composition of most substrates.

The XRD analysis results for UTGF-coated substrate A of different deposition times are shown as Figure 5b. The universal broad-base peaks of 2θ: 10–25° of all specimens indicated the amorphous behavior of collagen substrates and noises [43]. In comparison with the ICSD database (gold code: 44362), only the specimens of deposition times 51 and 77 s exhibited very weak diffraction peaks (111) of Au within the range of 2θ: 36.1–40.1°. The observed peaks were weak that detection by regular low glancing angle XRD equipment was difficult. The difficulty further verified the thin thickness of UTGF achieved by the HiPIMS system. Peaks other than (111)-preferred orientation, such as (200), (220), and (311) in the range of 2θ: 35–80°, were not able to be detected in this study. The XRD spectrum of 2θ: 36–40.5° for UTGF-coated substrate A of deposition times 26, 51, and 77 s is shown in Figure 5b. The short peaks with a wide base on 2θ: 38.5° were seen only on specimens of deposition times 51 and 77 s. The morphology of the peaks was similar to that in the literature [44], where Häupl et al. deposited 1.7 nm gold thin film on Cu substrate and observed the 2θ of the peak. More recently, Yan et al. [45] measured 2θ = 40° for AuNPs of 0.8–1 nm, and the results were more similar to our observation. Identical peaks of the preferred orientation of UTGF and Au bulk indicated the high energy status of HiPIMS Au plasma, resulting in higher cohesive force on the substrate surface.

### 3.3. Surface Roughness and Topography of UTGF-Deposited Specimens

The AFM study was based on substrate A rather than substrate B to ensure less confounding localization factors [46,47]. The 3D surface topography is shown in Figure 6. Bare collagen substrate had a larger range of peaks and valleys (81.7–1.2 nm), which was due to the light weight of the bare collagen membrane and poor attachment on the AFM holder. The characteristic wavy appearance due to dark and light band overlapping of collagen fibers could be observed. The appearance was blurry on bare substrate A and was enhanced on UTGF-coated specimens of deposition times 13, 26, and 51 s. The range of peaks and valleys was from 29.9 to −1.5 nm for the 51 s specimen and from 28.1 to −0.5 nm for the 77 s specimen, respectively. The flatter appearance of the UTGF-coated specimen for the deposition time of 77 s indicated decreased roughness due to the fusion of surface particles or island structures. Using the same thin film coating method and parameters on different substrates, Malinský et al. [48] discovered that no difference in UTGF thickness could be observed for film thickness less than 25 nm. Therefore, simultaneous UTGF deposition on glass substrate was obtained as substrate A. The film thickness and surface roughness were measured by AFM. The results are recorded in Table 3. The UTGF thickness measurement by cross section of SEM was not achieved due to the altered fine structure in paraffin embedding specimens and difficult preparation of collagen materials. However, the study on glass substrates confirmed the measurement.

The thickness of UTGF was 3–10 nm, and Au (111) peaks were noted by XRD only on films thicker than 7.98 nm, as shown in Figure 5b. The roughness of bare substrate was not well demonstrated consistently due to fragility. The Ra values of 13 and 26 s specimens were low and similar. Increased Ra was observed on specimens of 51 and 77 s with a linear trend. Similar to Ra, Rmax of the 13 and 26 s specimens were not significantly different. Rmax of 51 and 77 s were much larger than that of the 13 and 26 s specimens. However, no linear trend was observed by increasing the deposition time. According to gold thin film adhesion and growth theory [34,49], the observation suggested a granular to island structure growth of 13–26 s deposition time and percolation of 51–77 s. The difference in height for the UTGF-deposited and bare glass substrate represented film thickness. UTGF surface coverage percentage was calculated using the same method as in Figure 4. The relationship between deposition time and thickness or surface coverage of UTGF-coated collagen and glass substrates is plotted in Figure 7. The comparison of UTGF appearance and thickness on substrate A is listed in Table 4. The film thickness increases as the deposition time increases. UTGF of 10.37 nm on the glass substrate presented a discontinuous thin film with fluctuating microstructures, and part of the collagen substrates exposed with preserved characteristics and functions are clinically beneficial to skin wound recovery [1,31]. Under the same settings of HiPIMS deposition, the surface coverage of UTGF was lower on the glass substrate than on collagen. The coverage percentage difference between the glass and collagen substrate was as high as 22.05% at a deposition time of 30 s, which indicates that HiPIMS would be beneficial to bonding between AuNPs and collagen surface molecules [48].

### 3.4. Surface Chemistry and Secondary Structure of Collagen

The ATR-FTIR spectrum presented the substrate surface modification by HiPIMS plasma and UTGF deposition. The specific peaks of bands are described in Table 5 as follows: amide A by NH stretching and second amine, band at 3301 cm^−1^; amide B by CH_2_ stretching, band at 2960 cm^−1^; amide I by carbonyl stretching, band at 1630 cm^−1^; amide II by NH bend and CN stretching, band at 1547 cm^−1^; and amide III by CN stretching/NH deformation, band at 1239 cm^−1^. The peaks of CH_3_ asymmetric bending at 1451 cm^−1^ and CH_2_ twisting/wagging in the region between 1330–1343 cm^−1^ were also identified due to the stereochemistry of the pyrrolidine rings of proline and hydroxyproline [23,25]. The characteristics peaks were compatible with type I collagen in the literature [23,24,25,50]. No peak shifts were found, whereas decreased intensity was observed on the UTGF-deposited substrate. The observation showed that the substrate was preserved well after HiPIMS treatment, and no major change in the surface functional groups was observed. Some reports found slight shifts of amide I and II in terms of AuNPs conjugating with collagen colloids [10,51]. The current observation pointed out that the bonding of UTGF with collagen substrate was not achieved by occupying functional groups or altering chemical bonds of collagen. In the presence of oxygen ions, Au_2_O_3_ may form by overcoming the required energy of oxide formation of the deposited material [34]. However, in the HiPIMS system with Ar, the oxygen content is nearly zero, and no detectable corresponding peaks supported such oxide layer formation during HiPIMS UTGF deposition.

The ATR-FTIR spectrum of the bare and UTGF-coated substrates B of different deposition times was illustrated as Figure 8a. As the deposition time increased, the exposed collagen surface decreased. For UTGF-coated specimens of deposition times 24 and 30 s, surface functional groups of the collagen substrate decreased markedly. Compared with Figure 6, this finding shows that a UTGF coverage of more than 90% would seriously decrease the function of collagen substrates because a larger exposed collagen area would be more favorable in clinical applications [8,33].

The secondary structure of protein plays a major role in normal function, and any process on such materials should not alter the structure unless intended [52]. In proteins, the amide I vibration is affected not by different side chain structures but by the secondary structure of the backbone. Therefore, the amide I peak is most commonly used for secondary structure study [24]. As shown in Figure 8b, the secondary structure calculations using amide I band are α-helix (1651–1652 cm^−1^), β-sheet (1624–1625 cm^−1^), β-turn (1677–1696 cm^−1^), and aromatic ring vibration (1605–1606 cm^−1^) [40,41]. The intensity of signals decreases after HiPIMS UTGF deposition on substrate B, but most of the secondary structure remained α-helix, as in the natural environment [24,25]. The subgroup distribution of the secondary structure was also identical after HiPIMS UTGF deposition. The peak fitting analysis showed no bonding of the HiPIMS Au plasma species and atoms of the collagen backbone, such as C, N, and O. Moreover, the HiPIMS process preserved the chemical composition and physical structure of native collagen substrates. Thus, the biomedical application of such biomaterials is more practical.

The XPS system of this study could detect different elements status of 50 Å deep from the surface. Some authors recommended a binding energy (BE) of Au 4f_7/2_ for reference [53,54], and the common value was 84.0 eV. In this study, the peak of adventitious carbon, 284.5 eV, did not increase. The XPS spectrum of bare and UTGF-coated substrate B is shown in Figure 9a. The charge referenced to surface contamination is adventitious C1s (C-C peak at 284.8 eV), and the peaks of the shown spectrum are low. Deposition times of 24 and 96 s were selected for comparison. Surface elements of bare collagen were C, N, and O, which were compatible with the normal component of type I collagen [33,55]. In addition to the three basic elements, Au was detected on the UTGF-coated specimens. The surface element percentage of bare substrate B was measured as 61.7% for C, 20.9% for O, and 17.4% for N. The C/N atomic ratio was 3.55, which was higher than that of the collagen of mammals and birds [41] and indicated the enforced structure of collagen substrates by artificial processing [56,57].

The high-resolution spectrum by elements of 96 s UTGF-coated specimen is plotted in Figure 9b. The full width at half maximum (FWHM) of Au 4f_7/2_ was 1.09 eV, which represents high energy resolution. BE of Au 4f_7/2_ was 83.41 and 83.28 eV for the 24 and 96 s UTGF-coated specimens, respectively. Au 4f spin-orbit splitting was identical to reference [58,59], but the elevated BE of Au 4f_7/2_ peak should be observed for bonding with other elements or the unstable status of AuNPs [59]. The observed negative charge revealed no Au^+^ or Au^3+^ on the specimen, and the surface charge of Au is partially negative (Au^δ−^). Electron transfer from the collagen substrate to the Au particle surface with a strong interaction was proven. The result also excluded the oxidation of Au in HiPIMS UTGF deposition. FWHM of C 1 s was 1.7 eV, and BE of C 1 s peak was 284.88, 284.71, and 284.12 eV for bare, 24, and 96 s specimens, respectively. The slight cathodic polarization of C suggested C–O/C=O/C–N shifting to C–C on the surface [60]. FWHM of O 1 s was 1.66 eV, and BE of O 1 s peak was 530.93, 531.11, and 531.32 for bare, 24, and 96 s deposited collagen, respectively. Anodic polarization of O represented hydroxide or nitrate formation [61]. FWHM of N 1 s was 1.87 eV, and BE of N 1 s peak was 399.37, 399.48, and 399.30 eV for bare, 24, and 96 s deposited collagen, respectively. C-NH_2_ could be found in the main amino acid contents of collagen, such as glycine, proline, and hydroxyproline. The transient positive charge of the 24 s specimen suggested NH_3_ formation rather than metal nitride [62]. Overall, HiPIMS UTGF deposition on collagen substrate caused minimal change on the surface elements, and oxidation of Au, C, or N was not observed. Compared with the results of FTIR analysis, the main structure of collagen substrates was preserved in the HiPIMS system, and a slight chemical status shift was supposed to be related to the side chain reaction.

### 3.5. Surface Wettability

The results of the WCA study of bare and UTGF-coated substrate B are shown in Figure 10a. The water droplet on the post-pretreatment substrate was absorbed quickly, thus revealing the substrate’s hydrophilic behavior. The 24 and 96 s specimens showed transient hydrophobic behavior, which fitted the Wenzel wetting model of rough surface [63]. Water filled the exposed space between the gaps of UTGF and was absorbed rapidly by collagen within 1 s in all specimens. Collagen is a well-known hydrophilic material with high water absorption capacity [33]. Deformity of the surface with water-absorbed collagen could further alter the adherence of UTGF, and wettability was further changed. A box chart of WCA measurement of initial contact is presented in Figure 10b. The WCA of bare substrates could not be measured due to the immediate water droplet absorption. The average WCA of the substrates post-argon pretreatment was 38.6° with large diversity. The measured WCA of UTGF from the literature was not uniform [64,65]. Hydrophilicity was often observed in the presence of AuO on the surface due to certain film process and contamination, and hydrophobicity was rarely seen and always related to surface roughness and micro-nanostructure [66]. The WCAs of 6, 12, 24, 48, and 96 s specimens were 116.3, 120.0, 125.3, 127.4, and 124.8°, respectively. The transient hydrophobicity of UTGF by HiPIMS was related to the substrate surface topography and revealed the absence of gold oxidation or surface contamination.

## 4. Conclusions

It is worth knowing that HiPIMS was adopted for the first-time deposition on organic materials, namely collagen for this study. Collagen is assembled by three parallel polypeptide strands and plays a cardinal role in the skin extracellular matrix. The structure of collagen is important for normal strength, function, and biological behaviors, and denaturation of collagen substrates occurs in high temperature environments. UTGF deposition by the HiPIMS system on the two types of collagen substrates without damaging the gross morphology, surface functional groups, secondary structure, and element chemical status was successfully achieved and proven by surface analysis. The FTIR spectrum confirmed the unchanged amide A, B, I, II, III, and the pyrrolidine rings of proline and hydroxyproline, regardless of the deposition time. Amide I peak analysis also showed α-helix as the major secondary structure of collagen, as the bare substrates. The significant difference in Au atom percentage by EDS (0.98% in 77 s) and XPS (44.30% in 77 s) surface element analysis revealed that HiPIMS Au plasma only changed the composition of the superficial layer of substrates. High-resolution XPS spectrum confirmed electron transfer from substrate to Au by the presence of Au^δ−^. Through the combination of collagen and UTGF without destruction, both materials could demonstrate functions desired for biomedical purposes as well.

Based on the Volmer–Weber model, the initial growth of UTGF on collagen with the short deposition time of HiPIMS can preserve the interfibrillar space of substrates. With increasing deposition time from 6 to 30 s, the coverage of substrate surface area percentage significantly increased from 72.59 to 91.79%. The exposed interfibrillar space and porosity of collagen substrates were further decreased by UTGF deposition. Along with the high WCAs of UTGF, even in the shortest deposition time, the transient hydrophobicity and narrow exposed space were assumed to prohibit cell and bacterial adhesion in the composite of collagen and UTGF. The advantages of UTGF, such as degradation resistance, antibacterial activity, and antioxidant function, would benefit fragile collagen in biological environments such as exposed skin wounds.

## Figures and Tables

**Figure 1 nanomaterials-12-01627-f001:**
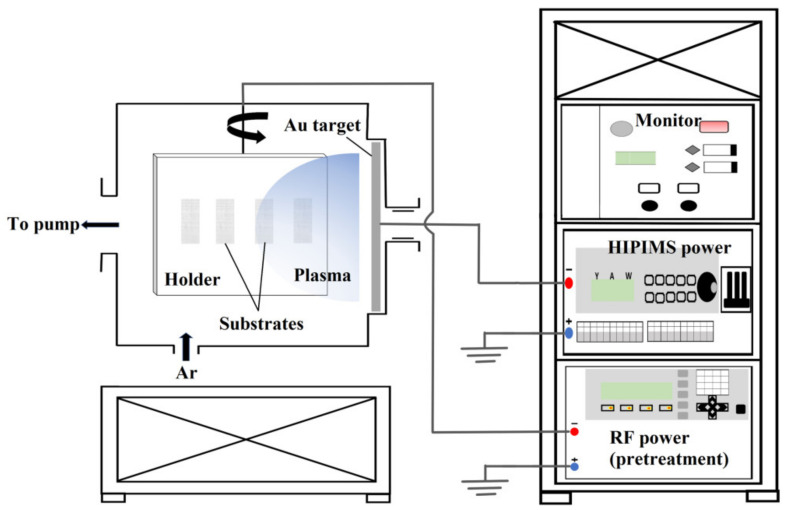
The schematic drawing of the HiPIMS system.

**Figure 2 nanomaterials-12-01627-f002:**
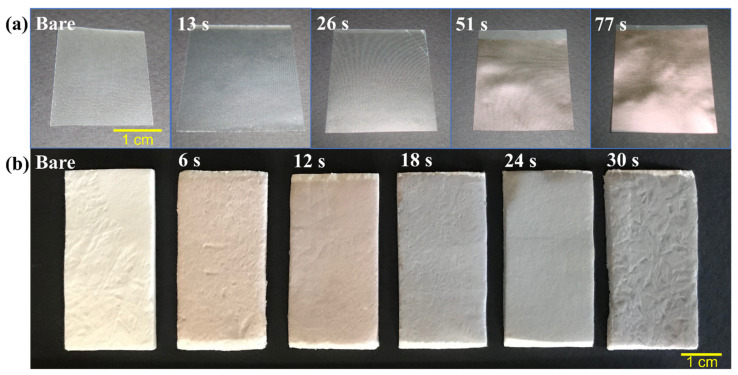
(**a**) Photos of substrate A of different ultrathin gold films (UTGF) deposition time, all specimens sized 2 cm × 2 cm. (**b**) Photos of substrate B of different UTGF deposition time, all specimens sized 4 cm × 2 cm.

**Figure 3 nanomaterials-12-01627-f003:**
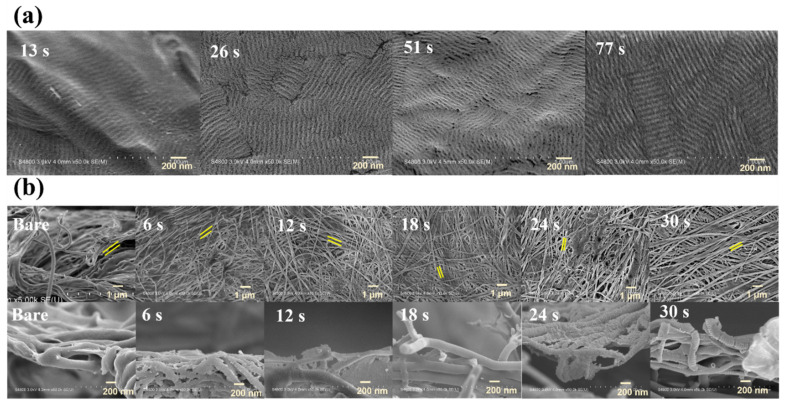
SEM images of UTGF-coated on (**a**) substrates A of deposition time 26 s and 51 s, and (**b**) substrates B of bare, UTGF-coated of different deposition time from a top view (upper row) and oblique view (lower row). Spaces between yellow lines indicate the interfibrillar space of collagen. Deposition of Pt was achieved on bare substrates. Morphology change of bare substrates in FE-SEM chamber was not observed before high electron energy application. Melting of fibrils was noted on the bare substrate B because of the high electron energy without the protection of UTGF. Pt deposition failed to completely protect the bare substrate due to relatively short deposition in the ion sputter.

**Figure 4 nanomaterials-12-01627-f004:**
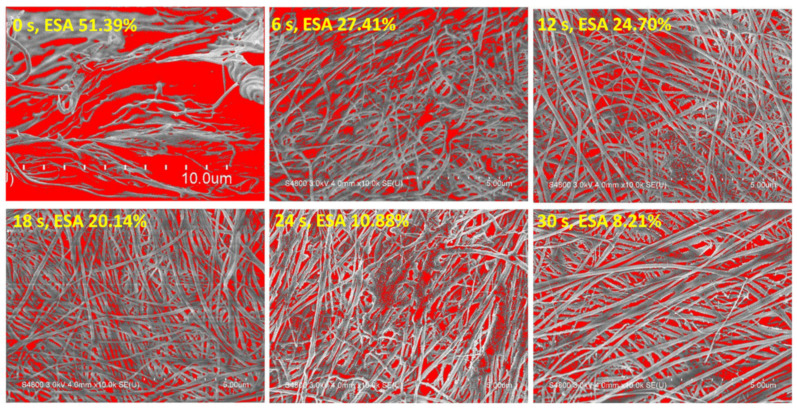
SEM images of UTGF-coated substrates B: red area indicated substrate exposure area process by the ImageJ software. The 0 s represented substrate post Ar pretreatment and HiPIMS vacuum without further deposition. The 6 s, 12 s, 18 s, 24 s and 30 s represented UTGF deposition time. ESA indicated substrate exposure area in percentage.

**Figure 5 nanomaterials-12-01627-f005:**
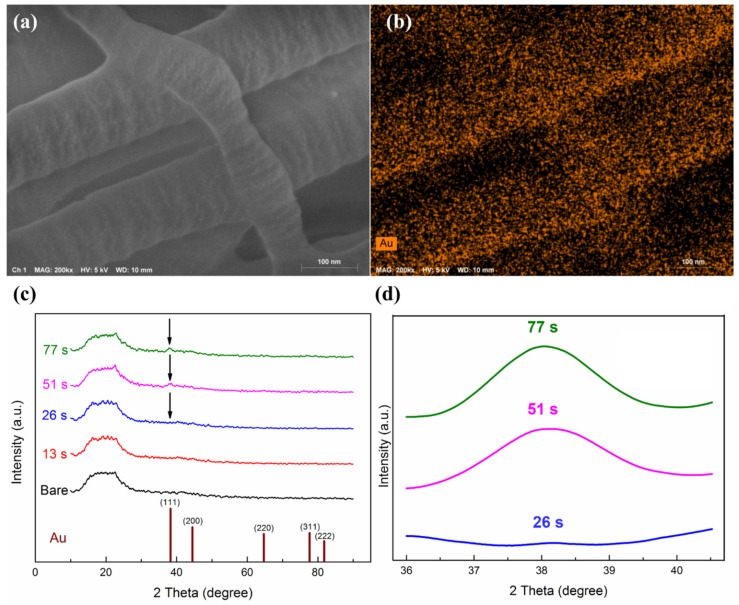
(**a**) SEM images of UTGF-coated substrate B at 6 s (magnification 200,000 times) and (**b**) EDS mapping of Au. (**c**) XRD spectrum of substrate A of different UTGF deposition time. Black arrows indicated peaks of Au (111). (**d**) High-resolution spectrum of 2θ: 36–40.5° on specimens of 26 s, 51 s and 77 s.

**Figure 6 nanomaterials-12-01627-f006:**
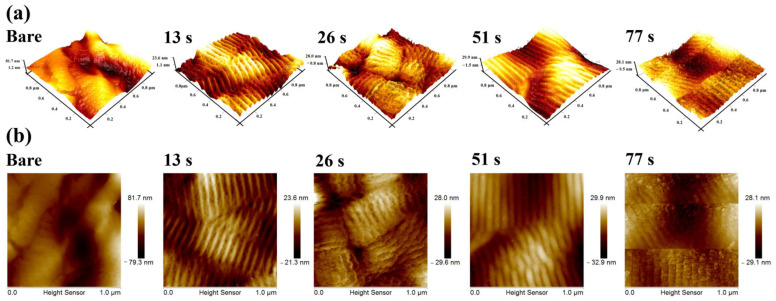
(**a**) AFM surface topography on substrate A: bare and UTGF films at deposition time of 13 s, 26 s, 51 s, and 77 s. (**b**) AFM color variation scale on substrate A: bare and UTGF films at deposition time of 13 s, 26 s, 51 s, and 77 s.

**Figure 7 nanomaterials-12-01627-f007:**
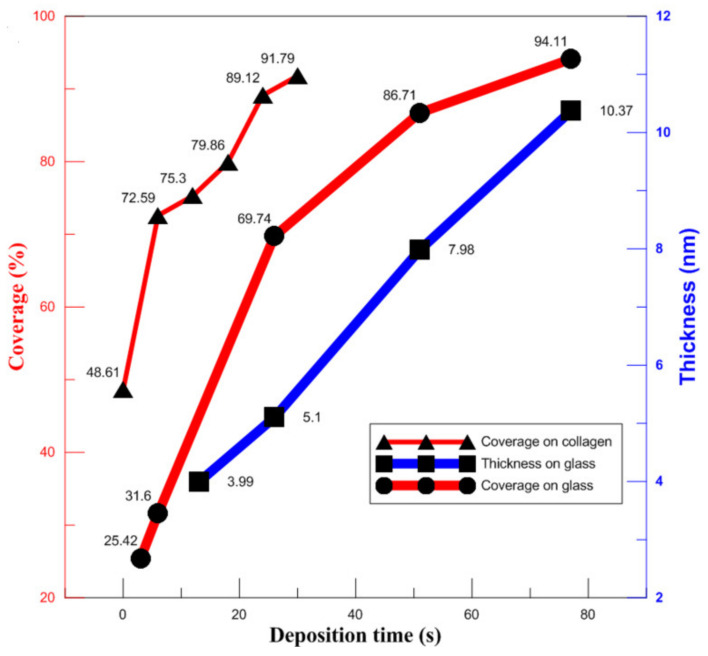
The plot of UTGF coverage rate versus UTGF deposition time. The thin red line (▲) indicated results of UTGF on substrate A. Thick red line (●) indicated results of UTGF on glass. The blue *Y*-axis represented the film thickness of UTGF. The blue line (■) indicated results of UTGF on glass simultaneously deposited with collagen substrates.

**Figure 8 nanomaterials-12-01627-f008:**
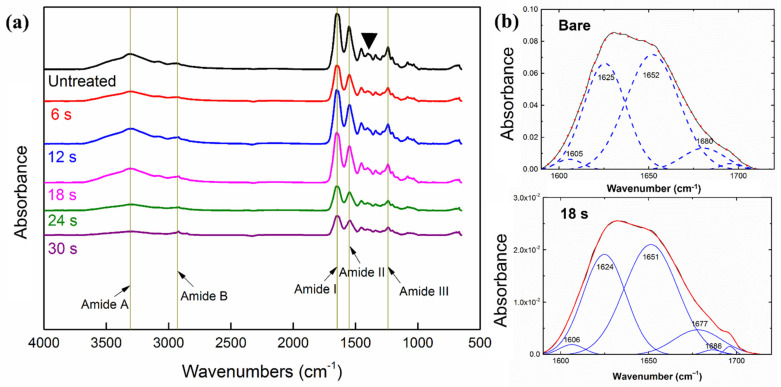
(**a**) ATR-FTIR spectrum of substrates B: bare and UTGF films at deposition time of 6 s, 12 s, 18 s, 24 s, and 30 s. Arrows and straight lines indicated amide groups. Arrowhead indicated peaks of pyrrolidine rings. (**b**) ATR-FTIR spectrum of substrates B: bare and UTGF deposition time 18 s. The blue lines were deconvolution curves and the numbers of different wavenumbers corresponded with the different secondary structures of protein. The red lines were fitted curves and the black lines were original data.

**Figure 9 nanomaterials-12-01627-f009:**
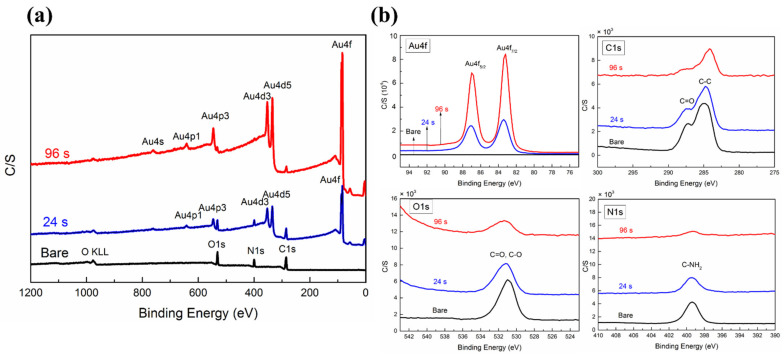
(**a**) Comparison of XPS broad spectrum of bare substrate B and specimens of UTGF deposition time of 24 s and 96 s. (**b**) The high-resolution spectrum of Au 4f, C 1 s, O 1 s, and N 1 s on bare and UTGF-coated substrate B.

**Figure 10 nanomaterials-12-01627-f010:**
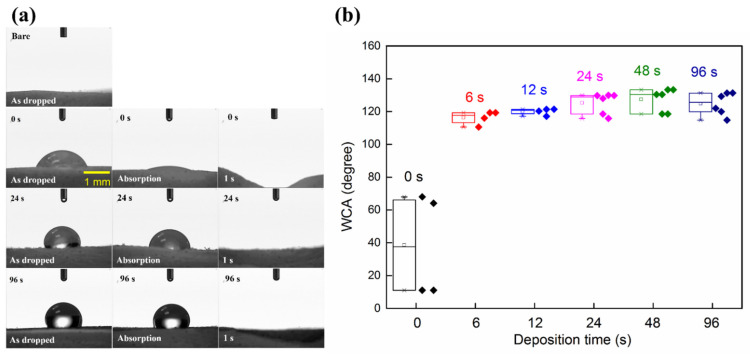
(**a**) Photos of WCA measurement on bare and UTGF-coated substrates B. 0 s represented substrate underwent vacuum treatment in HiPIMS chamber without sputtering. 24 s and 96 s represented HiPIMS gold deposition time. Bare substrate absorbed water immediately as dropped. Specimen of 0 s showed hydrophilicity and specimen with UTGF revealed hydrophobicity as dropped and during water absorption. Deformity of 0 s specimen was observed and all specimen absorbed water within 1 s. (**b**) Box chart of WCA on bare and UTGF-coated substrate B. The diamond dots represented a different measure of each sampling.

**Table 1 nanomaterials-12-01627-t001:** Process parameters for UTGF deposition on collagen substrates by HiPIMS. Target material: pure gold bonding target.

Parameters	Value
Targe voltage (V)	1391
Peak target current (A)	30
Peak current density (A/cm^2^)	0.106
Average target power (W)	133.1
Peak targer power (W)	41.7
Peak power density (W/cm^2^)	147.5
T_on_ (µs)	50
Pulse frequency (Hz)	100
Ar flow rate (sccm)	150
Working pressure (Pa)	0.54
Deposition time (s) for substrate A	13, 26, 51, 77
Deposition time (s) for substrate B (FTIR)	6, 12, 18, 24, 30
Deposition time (s) for substrate B (WCA, XPS)	0, 6, 12, 24, 48, 96

**Table 2 nanomaterials-12-01627-t002:** Surface element analysis by EDS and XPS on specimens of different deposition time.

**Energy-Dispersive X-ray Spectroscopy (EDS)**						
**UTGF Deposition Time (s)**		**Bare**	**13**	**26**	**51**	**77**
Atom (%)						
C		59.30	59.00	57.33	58.96	55.88
N		14.30	17.17	18.54	17.59	16.05
O		23.80	22.87	23.83	22.73	27.08
Au		0.00	0.10	0.31	0.72	0.98
**X-ray Photoelectron Spectroscopy (XPS)**						
**UTGF Deposition Time (s)**	**Bare**	**6**	**12**	**24**	**48**	**96**
Atom (%)						
C	61.70	57.00	57.60	55.70	46.30	40.40
N	17.40	14.20	12.10	11.90	7.90	4.80
O	20.90	19.10	19.50	17.70	14.20	10.50
Au	0.00	9.70	10.80	14.70	31.50	44.30

**Table 3 nanomaterials-12-01627-t003:** Surface roughness and film thickness of ultrathin gold films.

Deposition Time (s) *	13	26	51	77
Ra (nm)	1.74	1.68	2.40	4.17
Rmax (nm)	5.46	4.98	11.08	11.70
Film thickness (nm)	3.99	5.10	7.98	10.37

* The roughness of bare substrate was not shown due to inconsistent data. Ra and Rmax measurement was performed on the collagen substrate. Film thickness measurement was on the glass.

**Table 4 nanomaterials-12-01627-t004:** Morphology and UTGF thickness of substrate A of different deposition time.

Deposition Time (s)	Appearance	UTGF Thickness (nm)
13	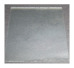	3.99
26	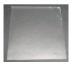	5.10
51	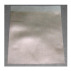	7.98
77	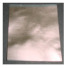	10.37

**Table 5 nanomaterials-12-01627-t005:** Peak analysis of FTIR spectrum from ultrathin gold films deposited on substrate B.

Peak Wavenumber (cm^−1^)	Functional Groups	Designed Bonding and Structure	References
3301	Amide A	NH stretching, secondary amine	[22]
3093		Overtone of Amide II	[23]
2960	Amide B	CH_2_ stretching	[21]
1630	Amide I	CO stretching	[22]
1547	Amide II	NH bend/CN stretching	[22]
1449	Pyrrolidine rings	CH_3_ asymmetric bending	[48]
1330–1343	Pyrrolidine rings	CH_2_ twisting/wagging	[48]
1239	Amide III	NH bend/CN stretching	[22]

## Data Availability

Not applicable.

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
