# Peer review of "Nanoarchitectonics for Ultrathin Gold Films Deposited on Collagen Fabric by High-Power Impulse Magnetron Sputtering"

_nanomaterials, 2022, doi:10.3390/nano12101627_

Round 1

Reviewer 1 Report

This is an interesting paper but there are some clarifications and explanations required.

line 32      Should “reserved” be “ preserved” ?

line 68      Clarify meaning of “electrical current of vacuum evaporation” and give reference,

line 70      “difficult for breakdown” – do you mean “difficult because of breakdown”? Clarify.

line 82      Omit comma after “higher”.

line 91     Replace “in fact” by “however”.

line 127    Clarify sentence “The performance of UTGF on collagen attaching to deposition time was then studied.” particularly the meaning of “attaching”.

line 133    “Bare substrates were deposited with platinum for the FE-SEM study” Does this have any effect on the collagen?

line 190   55 nm should be 51 nm. See also line 198.

line 235    “Melting of fibrils was noted on the bare substrate B because of the high electron energy without the protection of UTGF.” How much Pt was deposited and why does this not protect the collagen similarly to Au?

line 241    “The SEM images of different deposition times and collagen exposed ratio of the surface area are also shown in Figure 4 and processed with ImageJ software (Rasband, W.S., ImageJ, U.S. National Institutes of Health, Bethesda, Maryland, USA).” Rephrase this comment. The collagen at lower levels is still exposed, only at a lower level. Perhaps say something like “ratio of collagen on surface to collagen at deeper levels”.

line 263    “The results showed that sputtered Au ions only deposited on the surface of collagen substrates without deeper penetration via the porous structure or interfibrillar space.” Explain why this is so. Many of the sputtered particles are arriving at the substrate surface vertically and they would be expected to penetrate down holes until they reach an obstacle. Why is this not happening?

line 310    Table 2 – make clear which measurements are on glass and which on collagen.

Figures     Change the figures so that the different curves are distinguishable in black and white as well as colour.

Author Response

Dear Editor and Reviewers,

We would like to submit the original article entitled as “Nanoarchitectonics for Ultrathin Gold Films Deposited on Collagen Fabric for Skin Cell Recovery by High Power Impulse Magnetron Sputtering” to be reviewed for publication. Thank you for your letter dated on 5 May, 2022. The title of the article was revised by the Editor’s opinion. Based on the comments made by the reviewers, we have revised the manuscript and highlighted the changes and attached the clean and edited file, respectively. Reply to Reviewrs’ comments is also attached in a separate file. We would like to take this opportunity to express our sincere thanks to the reviewers who identified areas of our manuscript that needed corrections or modification.

All authors have read and approved the final manuscript. The manuscript is not under consideration by any other journal, and the work has not been previously published. Any persons who do not fulfill the requirements to be listed as authors but who contributed to the manuscript have been disclosed. Thank you very much for evaluating our manuscript and we look forward to hearing good news from you.

Yours Gratefully,

Corresponding author: Chia-Man Chou

Postal address: Department of Surgery, Taichung Veterans General Hospital, National Yang Ming Chiao Tung University, No. 1650, Sec. 4, Taiwan Boulevard, Situn District, Taichung 40705, Taiwan, R.O.C.

Tel: +886-4-23592525 ext.5182

Fax: +886-4-23741323

Reviewer 2 Report

  1. How has the thickness of the coating been confirmed and its adherence with the substrate?
  2. It is recommended to provide the thickness of the coating according to time and their physical appearance in a tabular form.
  3. In figure 6, please provide the color variation scale to identify the variation of the AFM surfaces
  4. Is the deposition time dependent on the deposition rate?, what are the influencing parameters?
  5. In figure 10, the wetting angle seems to be varying with time. Please provide the discussion about them.
  6. How the surface contamination can be identified and remedies are needs to be provided.

Author Response

(The authors gave the same response as above.)

Reviewer 3 Report

The submitted paper deals with the characterization of collagen fabric coated with gold particles at the nanoscale. The deposition was performed using high-power impulse magnetron sputtering and the process parameters were adjusted to balance the final properties and the substrate damaging. Several analytical techniques have been used to characterize the surfaces.

The topic is interesting and the study is properly organized. I would suggest the following minor changes before the publication.

- The abstract should be shortened.

- In the text, add the reference (e.g., on line 263) to XPS results listed in Table 1.

- Figure 5: Use letters to name each of the image/graph and add the related description in the label.

- Figure 10a: moving from left to right, what is the meaning of each image? Please, detail. In addition, in the photo on the right, the surfaces seem to be degraded. Discuss such behavior.

Author Response

(The authors gave the same response as above.)
